# Hematological Alterations after Cytoreductive Surgery and Hyperthermic Intraperitoneal Chemotherapy

**DOI:** 10.3390/jcm12134323

**Published:** 2023-06-27

**Authors:** Maria Consuelo Pintado, Inmaculada Lasa Unzúe, Remedios Gómez Sanz, Manuel Diez Alonso, Miguel A. Ortega, Melchor Álvarez de Mon, Emilio Nevado Losada, Alberto Gutierrez Calvo

**Affiliations:** 1Intensive Care Unit, Hospital Universitario Príncipe de Asturias, 28805 Alcalá de Henares, Spain; 2Department of Medicine and Medical Specialities, Faculty of Medicine and Health Sciences, University of Alcalá, 28801 Alcalá de Henares, Spain; 3Department of General and Digestive Surgery, General and Digestive Surgery, Hospital Universitario Príncipe de Asturias, 28805 Alcalá de Henares, Spain; 4Ramón y Cajal Institute of Sanitary Research (IRYCIS), 28034 Madrid, Spain; 5Inmune System Diseases-Rheumatology and Internal Medicine Service, Hospital Universitario Príncipe de Asturias (CIBEREHD), 28806 Alcalá de Henares, Spain

**Keywords:** cytoreduction, hyperthermic intraperitoneal, postoperative complications, prognosis

## Abstract

Background: Cytoreductive surgery (CRS) and hyperthermic intraperitoneal chemotherapy (HIPEC) have benefits for survival in some cancers with peritoneal metastasis. Hematologic toxicity described rate is 2 to 38%. Methods: Patients admitted to an intensive care unit (ICU) after CRS and HIPEC over 78 months. The data recorded were demographic characteristics, the severity of illness, complete blood samples, the type of cancer and extension, HIPEC drug and temperature, ICU and hospital stay and mortality, bleeding, and the need for transfusion of blood products. Results: Of the 96 patients included, 77.1% presented hematological complications: 8.3% leukopenia (<4000/mm^3^ leucocytes), 66.7% anemia (hemoglobin < 10 mg/dL), and 22.9% coagulopathy (INR < 1.5, or/and aPTT < 45 s, or/and platelet count < 100,000/mm^3^, or/and <100 mg/dL of serum fibrinogen). Leukopenia was higher in ovarian cancer or those treated with doxorubicin. Females with anemia, ovarian cancer, and those treated with cisplatin or doxorubicin had longer ICU stays. Bleeding complications were low-corrected in a conservative manner. The median ICU stay was 5 (4.0–5.0) days. The ICU mortality rate was 1.0%. Conclusions: In our study, 77.1% of patients treated with CRS and HIPEC developed hematological complications during the postoperative period; the majority of them were not severe and resolved spontaneously, without an effect on mortality or hospital stay.

## 1. Introduction

Cytoreductive surgery (CRS) and hyperthermic intraperitoneal chemotherapy (HIPEC) have demonstrated a decrease in peritoneal recurrence and a higher survival in some cancers with peritoneal metastasis.

In this technique, CRS consists of numerous surgical procedures depending on the extent of peritoneal tumor manifestation to achieve, if possible, no visible residual tumor. Followed by HIPEC, the chemotherapy agent is typically perfused at an inflow of above 43 °C to reach an intraperitoneal temperature of 41–42 °C [1,2].

Several postoperative complications have been described after CRS and HIPEC, related to surgery, hyperthermia, and/or chemotherapy. The described morbidity rates are 25% to 51%, and up to 21–30% patients develop severe postoperative complications, requiring surgical, endoscopic, or radiological reintervention or readmittance to the intensive care unit (ICU) [2,3,4]. The described mortality rate varies between 0 and 18% [5,6].

Hematologic toxicity is a complication that is not well established because few studies have been conducted on it, and in these, the results are very different, with percentages ranging from to 2 to 38% [6,7,8].

The purpose of this study is to define the clinical impact of alteration in coagulation and other hematological alterations that occur in immediate postoperative period after CRS and HIPEC during an ICU stay.

## 2. Materials and Methods

We included all consecutive patients who were admitted to our ICU after cytoreductive surgery and hyperthermic intraperitoneal chemotherapy, from 1 January 2013 to 30 June 2019. We excluded patients who refused to consent to the study. The study was approved by the Institutional Ethics and Clinical Trials Committee of University Hospital Príncipe de Asturias, and written informed consent was obtained from the patients. This study did not receive financial or logistical support from any institution outside the service or the hospital where the study was carried out; neither the researchers nor the subjects recruited in this study received any fees or material incentives for their participation.

The compiled data of every patient included the following: demographic characteristics, the severity of illness (measured by APACHE II [9]), complete blood samples (platelet count, international normalized ratio (INR), active partial thromboplastin time (aPTT), and serum fibrinogen) at admission and daily until ICU discharge, the type of cancer, Peritoneal Carcinomatosis Index (PCI) [1], HIPEC drug and temperature, ICU and hospital stay, ICU and hospital mortality, the presence of bleeding, and the need for the transfusion of blood products during surgery and ICU stay. When more than one value was available for a given day, the most abnormal value was recorded.

We defined leukopenia as a leukocyte count less than 4000/mm^3^ and anemia as a hemoglobin level equal to or less than 10 mg/dL. Due to the lack of a consensus regarding the precise definition of “coagulopathy”, we defined abnormal coagulation as an INR equal to or more than 1.5, or/and aPTT equal to or more than 45 s, or/and platelet count equal to or less than 100,000/mm^3^, or/and equal to or less than 100 mg/dL of serum fibrinogen. These criteria were applied in another study concerned with coagulation after CRS and HIPEC [7].

The management of patients during their stay in the ICU, including the transfusion of blood products, was conducted according to standard protocols. Venous thromboembolism prophylaxis with enoxaparin was used in all patients, except when a contraindication existed.

The normal distribution of variables was assessed using the Kolmogorov–Smirnov test. Quantitative variables with normal distribution are expressed as means ± S.D. and compared using the Student *t* test; non-normal distribution variables are shown as medians and interquartile ranges and compared using the Mann–Whitney test. Qualitative variables are shown as percentages and compared with the chi-square test.

Logistic regression models were used for the analysis of factors associated with a longer ICU stay (defined as more than 5 days). A univariate analysis of the main variables registered at ICU admission (age, sex, type of cancer, and APACHE II) and during ICU stay (the transfusion of blood products during ICU stay, and bleeding) was performed. For the analysis, some variables (APACHE II, ICU stay, and SOFA score) were dichotomized based on the median. Predictor variables that were statistically significant (set to *p* < 0.10) when evaluated individually were included in the forward stepwise multiple logistic regression analysis. The results are presented as odds ratios with 95% confidence intervals.

The level of statistical significance was set to a *p* ≤ 0.05, using bilateral contrast, and the results are expressed with a 95% confidence interval.

## 3. Results

During the study period, 97 patients were admitted to our ICU after CRS and HIPEC. One patient refused to consent to the study, so 96 patients were finally included in the study.

The mean age was 60.7 ± 9.7 years; 50% of patients were male.

In 78 patients (81.3%), the cancer had a digestive origin (43 (44.8%) patients had colorectal cancer, 33 (34.4%) had gastric cancer, and 2 (2.1%) had pseudomyxoma); in 18 patients (18.8%), the origin was ovarian, with a median peritoneal carcinomatosis index of 5.0 (1.2–10.7).

The median duration of surgery was 10.0 (8.5–11.0) h, during which 16 patients (16.7%) needed blood transfusion (a mean of two red blood cell concentrates).

The mean APACHE II [9] at ICU admission was 9.3 ± 3.9, with a mean SOFA score [10] of 1.0 (0.25–3.0). At ICU admission, 5 patients (5.2%) were under mechanical ventilation, and 11 (11.6%) had vasoactive support during or at ICU admission.

During admission, 74 patients (77.1%) presented some kind of hematological complications: 8 (8.3%) patients developed leukopenia, 64 (66.7%) developed anemia, and 22 (22.9%) developed coagulopathy.

Of the 74 patients included in the study, 4 patients presented bleeding through surgical drainages, 8 presented hematuria, and 1 presented digestive hemorrhage, but only 3 patients needed a transfusion of blood products. All bleeding complications were corrected in a conservative manner.

A total of 19 (19.8%) patients needed a transfusion of blood products due to progressive anemia without seeing active bleeding, probably related to the received chemotherapy treatment (only one patient had hematuria, one bled through peritoneal drainages, and one had low digestive hemorrhage).

The median ICU stay was 5 (4.0–5.0) days; the median hospitalization was 10.0 (9.0–13.0) days. One patient (1.0%) died during their ICU stay due to massive pulmonary embolism; none did so during their hospital stay after ICU admission.

There were no differences between the patients who developed coagulopathy or not, even in the development of bleeding complications (Table 1).

We found that female patients, those who had ovarian cancer, or those who were treated with cisplatin or doxorubicin developed anemia more frequently. Obviously, they received more transfusion of blood products during their ICU stay. Although they did not have a higher incidence of bleeding complications, these patients had a longer ICU stay (Table 2).

Anemia appears on day 3 postoperatively (Figure 1).

Leukopenia developed more frequently among patients with ovarian cancer or those that received doxorubicin as intraperitoneal chemotherapy (Table 3).

Logistic regression analysis was performed to assess the effect of clinical variables and hematological toxicity with a long ICU stay (defined as more than 5 days). Only age was found to have a trend towards a longer ICU stay (Table 4), and no factor was correlated with ICU mortality (Table 5).

## 4. Discussion

In our study, we found that hematological complications during the immediate postoperative period in the ICU stay were frequent (up to 77.1% of patients), although not severe, nor were they associated with the development of bleeding complications, a longer ICU stay, or ICU mortality, even in those who developed coagulopathy. The development of anemia was associated with a longer ICU stay and more transfusions of blood products.

We found that 66.7% of patients developed anemia, which was associated with a longer ICU stay and a greater need for transfusions, although without a statistically significant association with the presence of bleeding complications. A possible explanation for this fact is that in the group of patients with anemia, they received more blood transfusions and were kept in the ICU longer to ensure the absence of bleeding and/or the stability of hemoglobin levels after the blood transfusion.

As other authors have described, we found a progressive decrease in hemoglobin levels during the postoperative period, not due to bleeding complications [11,12,13,14]. There have been few studies focused on anemia after CRS and HIPEC. Falcon-Araña [11] found that there was only a slight decrease in hemoglobin rate during surgery—not related to visible bleeding or hemodilution—so they concluded that it may be explained by hemolysis due to hyperthermia or chemotherapy. Schmidt et al. [12] showed a reduction of 24% in hemoglobin rate during surgery, which recovered slowly during the postoperative period; in 6 of the 78 patients (7.7%) studied, a median of 0.3 (0.3–1.2) L of packed red blood cells was given in the postoperative period. Cooksley et al. [13] showed a statistically significant decrease in hemoglobin in the postoperative period with respect to the preoperative level, although only 15 patients (21%) required a transfusion of red blood cells in the postoperative period in the ICU. Somashekhar et al. [14] described an incidence of anemia (defined as <8.0 gr/dL) of 21.4% in 56 patients with peritoneal surface malignancies. Cripe et al. [15] described a median hemoglobin nadir of 8.6 gr/dL on day 5 postoperatively, including patients transfused intraoperatively and/or postoperatively, without a relation to prior chemotherapy exposure, in patients with recurrent ovarian carcinoma treated with CRS and HIPEC.

In our study, nearly 20% of patients needed a transfusion of red blood cells. The rates described varied from 7.7% postoperatively to 36% [7,12]. Interestingly, Nizri et al. [16] studied 231 patients with diffuse malignant peritoneal mesothelioma and 273 patients with pseudomyxoma peritonei who were operated on in their units (CRS and HIPEC). A total of 74% of them were transfused during perioperative period, with a median of two packed red blood cells. They found that red blood cell transfusion, even minimal, had a significant adverse effect on both short- and long-term outcomes in cytoreductive surgery, and, therefore, suggested focusing on transfusion reduction in these patients.

In these patients, thrombocytopenia is due to the cytotoxic drugs used and dilutional coagulopathy and blood loss [17]. Similar to our study, several studies have shown that the platelet count decreased to noncritical levels between postoperative days 1 and 3, and normalized at day 7 [7,18,19,20,21], without the need for platelet transfusion after surgery.

The describes rates of thrombocytopenia depend on the chosen cut-off point. The majority of studies have focused on the management of these patients with epidural analgesia, so the cut-off point taken is 100.000/mm^3^. Furthermore, most studies have defined coagulopathy as a mixture of platelet level and coagulation times (as a platelet count <100.000/mm^3^, INR > 1.5, or aPTT > 45 s); this makes it difficult to make a comparison between our study and those previously described. With this cut-off point, the described level of thrombocytopenia is around 17% [7], lower than in our study. Only two studies— one involving 18 patients treated with intravenous ifosfamide and HIPEC and the other involving 50 patients treated with CRS and HIPEC—described an incidence of 61.1 and 36%, respectively, with a cut-off of 150,000/mm^3^ as in our study [22,23]. Another described an incidence of thrombocytopenia (defined as lower than 140,000/mm^3^) of 37.9% in 235 CRS and HIPEC patients; these authors found that increased age (>60 years) and previous chemotherapy were independent risk factors associated with the development of thrombocytopenia [24]. Although we did not reach such a finding, severe thrombopenia (less than 50,000/mm^3^) is described in 0.6–11% of patients [7,19,25].

Coagulopathy was present in 22.9% of the patients included in our study. With respect to coagulation times, Cooksley et al. [13] described the postoperative critical care of 69 patients treated with CRS and HIPEC; they found a prolongation of coagulation laboratory tests (INR and aPTT) after the surgery, with a peak 24 h postoperatively, although the correction of abnormal coagulation with fresh frozen plasma was only indicated in one patient. Similar results have been described by other authors, who also reported that at postoperative day 3, INR and aPTT normalized without the need for fresh plasma transfusion [7,20,21,25,26].

Little is known about fibrinogen in these patients. Falcon Araña [11] demonstrated a decrease in fibrinogen during surgery, and it was associated with a greater need for blood product transfusions. Van Poucke et al. [18] showed a decrease in the concentration of plasmatic fibrinogen or a deterioration of its function, which recovered at day 7 and even improved compared to before surgery. We found that hypofibrinogenemia was associated with more transfusions of blood products during ICU admission; additionally, it was more frequent in patients with residual tumor after surgery.

Although several studies showed a significant increase in white blood cell count during the first 7 postoperative days, especially if a splenectomy had been performed [12,18,27,28,29], in our study, we found that eight patients (8.3%) developed leukopenia, which other authors also found. Horvath et al. [30], in their study involving 40 patients with pseudomyxoma peritonei treated with HIPEC, found that 31% of them developed leukopenia (<4000/mm^3^) between day 6 and day 7 postoperatively; all of them were treated with mitomycin. Hakeam et al. [22] found an incidence of 11.1% of leukopenia, which recovered spontaneously within 48 h without a need for filgrastim use. Wong et al. [24] found an incidence of leukopenia of 15.3% that recovered after 1–3 days and a 3.8% incidence of neutropenia (<2000/mm^3^), which was related to ICU stay; through a univariate analysis, they found that patients that received more intraoperative blood transfusions had a higher risk of postoperative leukopenia. Fefermann [31] found a 7% incidence of leukopenia (defined as <2000/mm^3^) in 242 patients treated with CRS and HIPEC with mitomycin C, with a median nadir on day 5 postoperatively, who had a significantly longer ICU stay with respect to patients who did not develop leukopenia (11 vs. 6 days, *p* < 0.05). Bartos et al. [4] found that only 2 of the 50 patients included in their study (4%) developed leukopenia; they included patients with peritoneal carcinomatosis of different origin. Wong et al. [32] found an incidence of 12.3% of leukopenia (defined as <4000/mm^3^) in 220 patients treated with CRS and HIPEC, with a median duration of leukopenia of 1 day (1–3 days). The difference in leukopenia rates between all these studies may be attributed in part to the difference in the protocol of intraperitoneal chemotherapy used, as well as the different definitions of leukopenia.

We also found a trend towards a longer ICU stay in patients who developed leukopenia, as already described by other authors [24,31].

We found a trend towards a higher incidence of leukopenia in the female sex. Lambert et al. [32] described the occurrence of neutropenia (defined as an absolute neutrophil count <1000/mm^3^) in 39% of patients treated with mitomycin C, with a median time to onset of 9 days after surgery and a median duration of 2 days; the factors related to neutropenia were the dose of mitomycin C and the female sex.

There are several limitations to our study, such as the lack of standardized anesthesia management and postoperative care. We included patients with different pathologies that received several chemotherapies. Another limitation was the lack of preoperative INR and aPTT values for all patients, although we anticipate that the majority of the patients would have shown normal test results the day before surgery. We did not measure the effect of surgery and HIPEC on renal or hepatic function, so we cannot assess the influence of renal or hepatic dysfunction on the observed hematological toxicity. We did not perform a multivariate analysis to see the factors associated with mortality or ICU stay.

The strength of our study is that it is the first involving patients with CRS and HIPEC which comprises all hematological toxicities—affecting platelets, red blood cells, leukopenia/neutropenia, fibrinogen, and coagulation times—and that it is centered on the immediate postoperative period in the ICU.

## 5. Conclusions

In conclusion, we found that 77.1% of patients treated with CRS and HIPEC developed hematological complications during the postoperative period; the majority of them were not severe and resolved spontaneously, without an effect on mortality or hospital stay. Only the development of anemia was associated with a longer ICU stay and more transfusions of blood products.

## Figures and Tables

**Figure 1 jcm-12-04323-f001:**
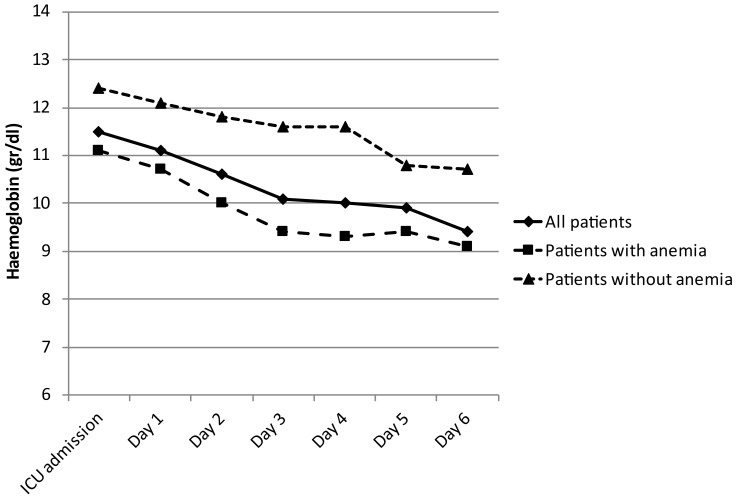
Hemoglobin evolution during first postoperative days.

**Table 1 jcm-12-04323-t001:** Characteristics of patients with coagulopathy.

	Patients with Coagulopathy (n = 22)	Patients without Coagulopathy (n = 74)	*p*
ICU admission			
Male gender (n and %)	13 (59.1%)	35 (52.7%)	0.33
Age, years (mean ± S.D.)	63.5 ± 8.9	59.88 ± 9.8	0.12
Type of cancer (n and %)			0.75
Digestive cancer	19 (86.4%)	59 (79.7%)	
Ovarian cancer	3 (13.6%)	15 (20.3%)	
Peritoneal Carcinomatosis Index (median and IR)	5.0 (2.0–7.25)	5.0 (1.0–11.0)	0.83
Intraperitoneal chemotherapy (n and %) *			
Mitomycin C	13 (59.1%)	49 (66.2%)	0.54
Cisplatin	8 (36.4%)	36 (48.6%)	0.31
Doxorubicin	3 (13.6%)	15 (20.3%)	0.75
Adriamycin	0 (0.0%)	1 (1.4%)	1.00
Oxaliplatin	6 (27.3%)	8 (10.8%)	0.05
Fluorouracil	4 (18.2%)	5 (6.8%)	0.20
Temperature (median and IR)	43.0 (43.0–43.0)	43.0 (42.0–43.0)	0.08
Completeness of Cytoreduction Score			0.10
0	22 (100.0%)	63 (88.7%)	
1–2	0 (0.0%)	8 (11.3%)	
APACHE II (mean ± S.D.)	10,0 ± 3.3	9.0 ± 4.1	0.32
Outcomes			
Transfusion of blood products during ICU stay (n and %)	2 (9.1%)	17 (23.0%)	0.22
Bleeding complications (n and %)	4 (18.2%) ^α^	8 (10.8%) ^β^	0.46
ICU stay (median and IR)	5.0 (4.0–5.0)	5.0 (4.0–5.0)	0.86
Hospital stay (median and IR)	10.0 (9.0–13.0)	10.0 (8.5.0–13.0)	0.49
ICU mortality (n and %)	0 (0.0%)	1 (1.4%)	1.00
Hospital mortality (n and %)	0 (0.0%)	1 (1.4%)	1.00

Abbreviations: n: number. IR: interquartile range. ICU: intensive care unit. * Patients received more than 1 chemotherapy drug; ^α^ hematuria (4 patients); ^β^ low digestive hemorrhage (1 patient), bleed through peritoneal drainages (4 patients), and hematuria (4 patients).

**Table 2 jcm-12-04323-t002:** Characteristics of patients with anemia.

	Patients with Anemia (n = 64)	Patients without Anemia (n = 32)	*p*
ICU admission			
Male gender (n and %)	27 (42.2%)	21 (65.6%)	0.03
Age, years (mean ± S.D.)	61.0 ± 9.9	60.0 ± 9.4	0.74
Type of cancer (n and %)			0.005
Digestive cancer	47 (60.3%)	31 (39.7%)	
Ovarian cancer	17 (94.4%)	1 (5.6%)	
Peritoneal Carcinomatosis Index (median and IR)	5.5 (2.0–11.0)	3.5 (1.0–10.2)	0.37
Temperature (median and IR)	43.0 (43.0–43.0)	43.0 (42.0–43.0)	0.06
Intraperitoneal chemotherapy (n and %) *			
Mitomycin C	37 (59.7%)	25 (40.3%)	0.05
Cisplatin	36 (81.8%)	8 (18.2%)	0.004
Doxorubicin	17 (94.4%)	1 (5.6%)	0.005
Adriamycin	0 (0.0%)	1 (100.0%)	0.33
Oxaliplatin	10 (71.4%)	4 (28.6%)	0.77
Fluorouracil	6 (66.7%)	3 (33.3%)	1.00
Completeness of Cytoreduction Score			0.36
0	56 (91.8%)	29 (90.6%)	
1–2	5 (8.2%)	3 (9.4%)	
APACHE II (mean ± S.D.)	9.5 ± 4.0	8.7 ± 3.8	0.25
Outcomes			
Transfusion of blood products during ICU stay (n and %)	18 (28.1%)	1 (3.1%)	0.003
Bleeding complications (n and %)	9 (14.1%)	3 (9.4%)	0.74
ICU stay (median and IR)	5.0 (4.0–5.0)	4.0 (3.0–5.0)	0.02
Hospital stay (median and IR)	10.5 (9.0–13.0)	9.0 (8.0–13.0)	0.059
ICU mortality (n and %)	0 (0.0%)	1 (3.1%)	0.33
Hospital mortality (n and %)	1 (3.1%)	0 (0%)	0.33

Abbreviations: n: number. IR: interquartile range. ICU: intensive care unit. * Some patients received more than 1 chemotherapy drug.

**Table 3 jcm-12-04323-t003:** Characteristics of patients with leukopenia.

	Patients with Leukopenia (n = 8)	Patients without Leukopenia (n = 88)	*p*
ICU admission			
Male gender (n and %)	1 (12.5%)	47 (53.4%)	0.059
Age, years (mean ± S.D.)	63.9 ± 11.5	60.4 ± 9.5	0.95
Type of cancer (n and %)			0.04
Digestive cancer	4 (5.1%)	74 (94.9%)	
Ovarian cancer	4 (22.2%)	14 (77.8%)	
Peritoneal Carcinomatosis Index (median and IR)	4.5 (1.5–7.5)	5.0 (1.2–11.0)	0.67
Temperature (median and IR)	43.0 (41.5–43.0)	43.0 (43.0–43.0)	0.95
Intraperitoneal chemotherapy (n and %) *			
Mitomycin C	3 (37.5%)	59 (67.0%)	0.13
Cisplatin	4 (50.0%)	40 (45.5%)	1.00
Doxorubicin	4 (50.0%)	14 (15.9%)	0.04
Adriamycin	0 (0.0%)	1 (1.1%)	1.00
Oxaliplatin	1 (12.5%)	13 (14.8%)	1.00
Fluorouracil	0 (0.0%)	9 (10.2%)	1.00
Completeness of Cytoreduction Score			0.66
0	8 (100.0%)	77 (90.6%)	
1–2	0 (0.0%)	8 (9.4%)	
APACHE II (mean ± S.D.)	9.1 ± 5.2	9.3 ± 3.8	
Outcomes			
Transfusion of blood products during ICU stay (n and %)	2 (25.0%)	17 (19.3%)	0.65
ICU stay (median and IR)	5.0 (5.0–5.7)	5.0 (4.0–5.0)	0.056
Hospital stay (median and IR)	9.0 (8.2–9.7)	10.0 (9.0–13.0)	0.14
ICU mortality (n and %)	0 (0.0%)	1 (1.1%)	1.00
Hospital mortality (n and %)	0 (0.0%)	1 (1.1%)	1.00

Abbreviations: n: number. IR: interquartile range ICU: intensive care unit. * Patients received more than 1 chemotherapy drug.

**Table 4 jcm-12-04323-t004:** Factors related to long ICU stay (more than 5 days).

	Odds Ratio (95% Confidence Interval)	*p*-Value
Univariate analysis		
Age, years	0.944 (0.882–1.010)	0.095
Male gender	3.841 (0.755–19.556)	0.105
APACHE II	0.879 (0.723–1.070)	0.879
SOFA	0.579 (0.298–1.122)	0.106
Hematological toxicity	1.061 (0.204–5.518)	0.944
Leukopenia	3.810 (0.644–22.523)	0.140
Anemia	1.018 (0.237–4.365)	0.981
Coagulopathy	0.000 (0.000–0.0)	0.998
Type of cancer	0.507 (0.059–4.336)	0.535
Transfusion of blood products during ICU stay	1.160 (0.221–6.091)	0.861
Presence of bleeding	0.852 (0.097–7.487)	0.885
Multivariate analysis		
Age	0.944 (0.882–1.010)	0.095

**Table 5 jcm-12-04323-t005:** Factors related to ICU mortality.

	Odds Ratio (95% Confidence Interval)	*p*-Value
Univariate analysis		
Age, years	0.938 (0.781–1.128)	0.498
Male gender	0.000 (0.000–0.0)	0.998
APACHE II	0.840 (0.463–1.521)	0.564
SOFA	1.018 (0.929–1.115)	0.702
Hematological toxicity	0.000 (0.000–0.0)	0.997
Leukopenia	0.000 (0.000–0.0)	0.999
Anemia	0.000 (0.000–0.0)	0.997
Coagulopathy	0.000 (0.000–0.0)	0.998
Type of cancer	0.000 (0.000–0.0)	0.999
Transfusion of blood products during ICU stay	8.975 (0.000–0.0)	0.997
Presence of bleeding	0.000 (0.000–0.0)	0.999
Multivariate analysis		
Age	0.938 (0.781–1.128)	0.498

## Data Availability

Clinical data are available upon reasonable request.

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
