# Peer review of "Hematological Alterations after Cytoreductive Surgery and Hyperthermic Intraperitoneal Chemotherapy"

_jcm, 2023, doi:10.3390/jcm12134323_

Round 1

Reviewer 1 Report

The concept of the study is in principle laudable, but the study population is very heterogeneous so in my opinion the value of the reported data is limited, I do not have the impression I learned much compared to earlier published clinical experience from higher volume CRS and HIPEC centers. Still one could consider publishing these real-world data.

I see no major flaws, relevant literature is cited, discussion is appropriate. There are some minor language issues that should get corrected.

there are some minor issues with the language

Author Response

We appreciate the opportunity to address the Reviewers’ comments to be able to improve the quality of our manuscript. 

We agree with the reviewer that the populations is very heterogeneous (just like in real life), so the value of our data is limited.

We have revised and corrected the spelling of some words, and sentences.

Reviewer 2 Report

This article showed the hematological complications after CRS and HIPEC. Authors emphasize that this result was not severe toxicity. However, the safety of this treatment strategy seems to be several problems in this manuscript. The conclusion may be not reasonable from the clinical data of this study.

Major

1.     In this study, univariable analysis comparing each hematological toxicity (yes / no) was only performed. However, independent risk factors for long hospital/ICU stay, mortality, postoperative complication should be identified. I think, the evaluation for the factor concerning severe clinical outcome is required if authors emphasize the safety (no severity) of hematological toxicity.

2.     In Table 2, why one case without anemia require transfusion?

3.     The cases with anemia require long ICU stay. These cases had no severe events?

4.     In this study, renal disorder after HIPEC was evaluated? Renal disorder affects the hematological toxicity. Please discuss more about renal disorder after HIPEC.

5.      

Minor

1. In abstract and so on, spell miss: “haematological” is correct? In the title, authors select “hematological”.

none

Author Response

We appreciate the opportunity to address the Reviewer’ comments to be able to improve the quality of our manuscript. Please, find below, item-by-item, responses to the Reviewer’ comments, which are included verbatim. All page and paragraph numbers refer to their locations in the revised manuscript.

Responses to Reviewer #2:

Major

  1. In this study, univariable analysis comparing each hematological toxicity (yes / no) was only performed. However, independent risk factors for long hospital/ICU stay, mortality, postoperative complication should be identified. I think, the evaluation for the factor concerning severe clinical outcome is required if authors emphasize the safety (no severity) of hematological toxicity.

We agree with the reviewer that only a univariate analysis was performed, so we cannot assess the safety of haematological toxicity. Thus, we have carried out a multivariate study to assess the factors associated with a longer stay in the ICU and mortality in the ICU, which we have added to the results. So, in the revised version we added in Results (last paragraph): “Logistic regression analysis was performed to assess the effect of clinical variables and hematological toxicity with long ICU stay (defined as more than 5 days) and ICU stay. Only age was found to have a trend towards a longer ICU stay (table 4), no factor was correlated with ICU mortality (table 5)”. With the addition of the two tables (tables 4 and 5).

Table 4. Factors related to long ICU stay (more than 5 days)

Odds Ratio (95% Confidence Interval)

P-Value

Univariate analysis

Age, years

0.944 (0.882-1.010)

0.095

Male gender

3.841 (0.755-19.556)

0.105

APACHE II

0.879 (0.723-1.070)

0.879

SOFA

0.579 (0.298-1.122)

0.106

Hematological toxicity

1.061 (0.204-5.518)

0.944

Leukopenia

3.810 (0.644-22.523)

0.140

Anemia

1.018 (0.237-4.365)

0.981

Coagulopathy

0.000 (0.000-.)

0.998

Type of cancer

0.507 (0.059-4.336)

0.535

Transfusion of blood products during ICU stay

1.160 (0.221-6.091)

0.861

Presence of bleeding

0.852 (0.097-7.487)

0.885

Multivariate analysis

Age

0.944 (0.882-1.010)

0.095

Table 5. Factors related to ICU mortality

Odds Ratio (95% Confidence Interval)

P-Value

Univariate analysis

Age, years

0.938 (0.781-1.128)

0.498

Male gender

0.000 (0.000-.)

0.998

APACHE II

0.840 (0.463-1.521)

0.564

SOFA

1.018 (0.929-1.115)

0.702

Hematological toxicity

0.000 (0.000-.)

0.997

Leukopenia

0.000 (0.000-.)

0.999

Anemia

0.000 (0.000-.)

0.997

Coagulopathy

0.000 (0.000-.)

0.998

Type of cancer

0.000 (0.000-.)

0.999

Transfusions of blood products during ICU stay

8.975 (0.000-.)

0.997

Presence of bleeding

0.000 (0.000-.)

0.999

Multivariate analysis

Age

0.938 (0.781-1.128)

0.498

And in Material and Methods: “Logistic regression models were used for the analysis of factors associated with longer ICU stay (defined as more than 5 days). Univariate analysis of main variables registered at ICU admission (age, sex, type of cancer, APACHE II) and during ICU stay (transfusion of blood products during ICU stay, bleeding) was performed. For the analysis, some variables (APACHE II, ICU stay, SOFA score) were dichotomized based on the median. Predictor variables that were statistically significant (set to p <.10) when evaluated individually were included in the forward stepwise multiple logistic regression analysis. The results are presented as odds ratios with 95% confidence intervals”.

And in Discussion (first paragraph): “In our study, we found that haematological complications during immediate postoperative period in ICU stay are frequent (up to 77.1% of patients), although not severe nor associated with development of bleeding complications, longer ICU stay nor ICU mortality, even those who developed coagulopathy”.

  1. In Table 2, why one case without anemia require transfusion?

We agree with the reviewer that it is rare that a patient without anemia would be transfused. The explanation for this fact is that a patient began to become hypotensive and the output from the surgical drains changed from serous to hematic; so, he was transfused by the doctor on duty without waiting for the result of the blood count. When the result of the complete blood count arrived, although the hemoglobin had dropped compared to the previous one, it did not meet the anemia criteria chosen in this study.

  1. The cases with anemia require long ICU stay. These cases had no severe events?

We did not find that patients with anemia had more bleeding complications nor severe events. A possible explanation for this fact is that in the group of patients with anemia, they received more blood transfusions and were kept in the ICU longer to ensure the absence of bleeding and/or the stability of the hemoglobin level after the blood transfusion. So, in the revised version we added in Discussion (2nd paragraph): “We found that 66.7% of patients developed anaemia, which is associated to a higher ICU stay and higher need of transfusions; although without a statistically significant as-sociation with the presence of bleeding complications. A possible explanation for this fact is that in the group of patients with anemia, they received more blood transfusions and were kept in the ICU longer to ensure the absence of bleeding and/or the stability of the hemoglobin level after the blood transfusion”.

  1. In this study, renal disorder after HIPEC was evaluated? Renal disorder affects the hematological toxicity. Please discuss more about renal disorder after HIPEC.

We agree with you that renal and hepatic disorder after HIPEC may affects hematological toxicity, but we did not measure it and it is a limitation of our study. So, in the revised version we added in Discussion (limitations paragraph): “We did not measure the effect of surgery and HIPEC on renal or hepatic function, so we cannot assess the influence of renal or hepatic dysfunction on the observed haematological toxicity”.

Minor

  1. In abstract and so on, spell miss: “hematological” is correct? In the title, authors select “hematological”.

Thanks for the observation. We have revised and corrected the spelling of the word “hematological”.

Round 2

Reviewer 1 Report

the authors have done a decent job

Reviewer 2 Report

Thank you for your response. I have no additional comment.